# A Cross-Sectional Study of Foot Growth and Its Correlation with Anthropometric Parameters in a Representative Cohort of Schoolchildren from Southern Spain

**DOI:** 10.3390/ijerph18084031

**Published:** 2021-04-12

**Authors:** María Luisa González-Elena, Emilio Fernández-Espejo, Aurora Castro-Méndez, María Dolores Guerra-Martín, Antonio Córdoba-Fernández

**Affiliations:** 1Departamento de Podología, Universidad de Sevilla, 41009 Sevilla, Spain; maruchi1@us.es (M.L.G.-E.); auroracastro@us.es (A.C.-M.); 2Reial Acadèmia de Medicina de Catalunya, 08001 Barcelona, Spain; efespejo@us.es; 3Departamento de Enfermería, Universidad de Sevilla, 41009 Sevilla, Spain; guema@us.es

**Keywords:** foot size, puberty, adolescents, pubertal onset, height, growth velocity

## Abstract

*Background*: The relationship between growth of the foot and other anthropometric parameters during body development until puberty has been scarcely studied. Some studies propose that growth of the foot in length may be an early index of puberty. The objective of this cross-sectional study was to analyze the relationship between the growth of the foot in length and width with other anthropometric parameters, in prepubertal and early pubertal schoolchildren (Tanner stage II). *Methods*: Using an instrument that was designed and calibrated for this purpose, maximum foot length, width and height were obtained in 1005 schoolchildren. *Results*: The findings indicate that the age of onset of pubertal foot growth spur was 7–8 years in girls, and 8–9 years in boys. Growth in foot length stabilized in both sexes after 12 years of age. In boys, a strong correlation was found between height and foot length (r = 0.884; *p* < 0.047), and between body mass index (BMI) and forefoot width at 12 years of age (r = 0.935; *p* < 0.020). A strong correlation was found between height and forefoot width at 6 years in girls (r = 0.719; *p* < 0.001), as well as between BMI and metatarsal width in 10 years-old girls (r = 0.812; *p* <0.001). *Conclusions*: The average increase in foot length and width that precedes the onset of Tanner’s stage II in both girls and boys can be considered as a useful biological indicator of the onset of puberty.

## 1. Introduction

The body growth that occurs during childhood and adolescence is a multifactorial process, where multiple variables are known to be involved. For this reason, several operational schemes for the study of body growth have been proposed. The biological organization of human growth is mainly related to age, gender, ethnicity, state of pubertal development, and the nutritional status of the individual. Taken together, these variables are considered as a multiple biomarker of the health status of a subject [1]. One of the most comprehensive schemes for the study of the biological organization of human growth was that proposed by Vandervael. This author suggests that the maximum development of lower limbs occurs before puberty, and bone development in limbs takes place before muscle maturation [2]. Ossification and bone maturation in the foot occurs before that in the long bones of the limbs.

Many cross-sectional studies suggest that the peak of foot growth takes place just before puberty. This process would be an early pubertal event that occurs before reaching the maximum body height [3,4]. Even though critical changes in feet are believed to occur during pre-school development, considerable changes also take place during school age and until late adolescence [5]. Some studies in early adolescents have shown that foot growth is faster in adolescents with lower levels of gonadal hormones, and that average annual percentage increase in foot length in prepubescent can be considered as an early index the transition to puberty [6,7]. Annual foot length is associated with annual increase in height, weight, and the Tanner’s stage in boys and premenarchal girls. Although changes in foot dimensions that occur early during puberty are easy to demonstrate, this parameter has been scarcely used to analyze human growth, or to study its correlation with parameters such as body size, weight, or body mass index (BMI). Its relationship with other anthropometric and hormonal parameters is yet unclear. To date, there are few studies on foot growth in schoolchildren that would allow us to follow up more accurately how the foot grows at each stage of development.

Anthropometric studies are needed to analyze the phenomena of biological maturation throughout the development of the individual. The main objective of this cross-sectional study was to analyze the relationship between foot growth and several anthropometric parameters in a broad and representative cohort of schoolchildren from southern Spain.

## 2. Materials and Methods

### 2.1. Setting

A cross-sectional and observational study was carried out with the aim of analyzing, comparing and correlating foot growth and other anthropometric variables in schoolchildren. The study was carried out from January 2015 to June 2019 within the framework of a school health program. The study was approved by The Research Ethical Committee of the Universidad de Sevilla (ref. 0711-2013). Once the parents or tutors were informed of the investigation, informed consent forms were obtained for the participation of the schoolchildren in this study. This cross-sectional study was carried out according to established criteria included in Strengthening the Reporting of Observational Studies in Epidemiology (STROBE). 

### 2.2. Participants

This study was performed through a purposive sampling and using the geographical proximity of participants to the Faculty of Nursing, Physioterapy and Podiatry of the Universidad de Sevilla as a selection criterion. Six centers located in the city of Sevilla (Spain) were selected. The randomized sample included one Early Childhood Education Center, four Primary Education Centers and one Secundary Education Center. The inclusion criterion was subjects’ age ranging from 3 to 12 years, and the exclusion criteria were the presence of foot deformities or previous pathology on lower limbs. One thousand fifty four subjects were initially recruited, and 49 subjects were discarded because they did not meet all research criteria. The final size of the cohort was 1005 schoolchildren. 

### 2.3. Measumerents

The longest foot was selected, and maximum foot length, width and height were measured. For making the measurements traceable, the participants stood barefoot, with both feet at the same height and their knees in extended position. The length was obtained directly by a transparency sheet through the grid scale. The width was obtained from the X and Y 115 coordinates of each point using Access 11.0 (Microsoft Office 2010) software. Subsequently a calibrated template (Andalusian Centre of Metrology; model CAM-V-00014-GRID-POD-03) was used to check the footwear. All the measuring instruments were designed and evaluated by the Andalusian Centre of Metrology [8]. Others demographic data as ethnicity, age and gender were registered too.

The inter-observer reliability of the researchers and the measurements were calculated with an interval of 1 week. Inter-observer intraclass correlation coefficient (ICC) was 0.99 for FL, 0.98 for FW, 0.99 for footwear length (FWL), and 0.99 for footwear width (FWW; level of confidence, 95%; Cronbach’s α). The coefficient of variation (CV) was 13.15% for FL, 12.59% for FW, 13.08% for FWL, and 9.74% for FWW. Inter-observer relative technical error of measurement (TEM %) was 0.20, 0.07, 0.21, and 0.06 respectively.

### 2.4. Statistical Analysis

Regarding descriptive statistics of dichotomous variables, absolute and relative frequency were calculated. As for quantitative variables, we calculated mean, standard deviation of the mean, and percentiles. The normality of the distribution (Kolmogorov-Smirnov test) and the comparison between groups were evaluated (the Chi-square test for dichotomous factors, and the Student t-test for quantitative variables). The Mann–Whitney U-test was used if needed. As for correlations between quantitative variables, the Pearson and Spearman tests were used for parametric and non-parametric distributions, respectively. All data was given as average ± standard deviation (SD). The foot anthropometric variables in the different age groups in males and females´ groups are illustrated. Significance was set at *p* < 0.05. Data were analyzed with the IBM^®^ SPSS Statistics statistical program version 25 for Windows 10 (IBM Corp, Armonk, New York, NY, USA).

## 3. Results

One thousand five schoolchildren were evaluated, of whom 507 were boys and 498 girls, with an average age of 7.53 ± 3.11 years (mean ± SD). Regarding ethnic origin, 96.2% of the schoolchildren were Caucasian, 1.6% South American, 1.8% African and 0.4% Asian. Boys were taller than girls at all ages, except at 10 and 11 years of age. Distribution according to age group and gender is shown in Table 1.

The left foot was found to be longer than the right one in 71% of schoolchildren (209.32 ± 31.42 mm versus 205.38 ± 30.95 mm), although differences were not significant. The average foot length of the boys was similar to that of the girls. Metatarsal width in boys was greater for all ages except at 3 years, where girls’ foot was significantly wider (Table 2).

The pubertal growth spurt in stature occurred in 6–7 year-old boys, and in 7–8 year-old girls. The age at pubertal foot length growth spur onset occurred between 8–9 years in boys and at 7–8 years in girls (Figure 1 and Figure 2).

In our cohort of children, a strong correlation was found between height and foot length (r = 0.884; *p* < 0.047), and between BMI and forefoot width at 12 years of age (r = 0.935; *p* < 0.020). In girls, a strong correlation was found between height and forefoot width at 6 years of age (r = 0.719; *p* < 0.001), and between BMI and metatarsal width at 10 years of age (r = 0.812; *p* < 0.001) (Table 3 and Table 4).

## 4. Discussion

Several authors state that each child has his/her “maturational tempus”, and that the pubertal growth spur, although it can influence growth in height during maturational development, does not seem to influence the final height of the individual [9,10,11,12,13,14]. In our cohort, the average height value was higher in the boys than in the girls at all ages, with significant differences at 7 and 9 years of age. However, at 10 and 11 years of age, no significant differences between boys and girls were found, although average height in girls was higher than that in boys. The same trend has been observed in other cross-sectional studies in school populations in northern Spain, where it was observed that girls had a higher average height than boys from 8 to 11 years of age, although differences were not significant [9,10,11,12,13,14]. In our study, the peak of growth in stature occurred in girls at 9–10 years. In our boys´ population, this first peak of growth in height or final height could not be established since only schoolchildren up to 12 years of age were evaluated.

Regarding foot length, in the present study the average foot length of the boys was found to be not significantly greater than that of the girls, except at 8 years of age. This observation is consistent with that reported in most studies carried out in the European population [15,16,17]. Several studies reveal ethnic anthropometric differences in the development of foot shape. Asiatic populations such as Japanese children have a wider foot length compared to Caucasoid, Africans and Australoid populations [18,19,20,21]. After comparing our results with similar studies conducted in Asian schoolchildren, it is observed that this population present shorter foot length at any age and gender. Another observation is that Asiatic boys´ feet have greater length and width relative to girls, and differences are significant from 7 through 12 years of age [19,21]. In Chinese schoolchildren the smallest differences for all foot measurements occurred at 7–9 years in both sexes, in contrast to our study that it occurred at 10–11 years [21]. However, when the results of our study and similar studies carried out in our country or Europe are compared, it is observed that, aside some differences, the development in length is very similar [14,15,16]. We believe that the differences between European and Asiatic schoolchildren may be due to racial factors rather than to methods of measurement or sample size.

As in previous studies, we also observed that the peak of growth in foot length was prior to the peak of growth in height, specifically in girls from 7 to 9 years of age, and in boys from 8 to 10 years (Figure 1). In accordance with other studies, we have observed that in females, the peak of longitudinal growth of the foot took place at 7–8 years, and it was prior to the peak of growth in height that occurs at 9–10 years [6,7,14]. Our results coincide with other studies that establish that the abrupt increase in foot length coincides in girls with Tanner’s stage II (7-8 years), and it occurs at 10–15 years in boys. These growth peaks indicate the onset of puberty [3,6,7]. In this study, we have not been able to establish exactly when the peak of growth in stature occurs in boys because we have only assessed schoolchildren up to 12 years.

Other authors have proposed that the sudden longitudinal growth of the foot occurs in preadolescence, and it is prior to the peak of maximum growth in height. Thus, a rapid longitudinal growth of the foot in preadolescence could be used as an indicator of the onset of puberty [4,6,7]. This theory is also proposed by other authors who also establish that this peak of growth in foot length occurs approximately in girls at 10.5 year of age, and one year later in boys. In such a way that this peak of growth in foot length occurs one year before the peak of growth in height in girls, and 2.5 years before the peak of growth in boys, up to 12 years [6].

The available evidence shows that, in children and teenagers, overweight has an impact on foot structures, with changes in the anatomical structures, abnormal distribution of the plantar pressure and different growth pattern than that of normal-weight children [22,23]. In this study, we have observed that weight and BMI values are somewhat higher than those reported in other studies carried out in schoolchildren in northern Spain [9,10]. The gradual increase in weight and BMI that occurs from 3 to 6 years in the mentioned studies, it occurred from 7 years of age on in boys, and from 8 years of age on in girls in our population. With respect to other cross-sectional studies that have been carried out in Spain, we observed that the increase in BMI occurred in more gradual and progressive way than that observed in the schoolchildren in our study [9]. This observation could be related to the high rate of child overweight registered in Andalusia compared to other regions of Spain, which reached a rate of 33.40% in 2017, five percentage points higher than the Spain’s mean [24].

Regarding the final time-point of the longitudinal growth of the foot, we have observed that growth slows down in both sexes after 12 years. Most studies report that the growth rate of foot length decreased after 12–13 years in adolescent girls continued to increase up to the age of 15 years in adolescent boys. Feet of girls stop growing around 12–13 years of age, and in boys, around 13–15 years [4,5,6,7,8,14,15]. In studies with cohorts of Asiatic schoolchildren, this event seems to end somewhat later, lasting up to 15 years in girls, and up to 16 years in boys [18,21]. Given the final age of the boys in our study, we have not been able to confirm this observation. Finally, in accordance with other studies with Spanish schoolchildren, we have observed that foot growth stabilizes earlier in girls than in boys [14].

Most of the studies carried out in European populations of schoolchildren report that the growth of the feet occurs in a very similar way in both sexes. From this age and up to 12 years, the dimensions of the feet begin to differ significantly between gender until the end of puberty [14,16,25]. In our sample, we have observed that the average value of growth in foot length between 3 and 7 years was higher in boys, and from 7 to 9 years of age this trend is reversed in favor of girls, although without significant differences. From 9 to 11 years, growth in foot length becomes similar in both sexes. Delgado et al. found that children have a greater average foot length from 6 to 12 years old, and that the greatest differences regarding the growth of the feet between both genders occur at 8 and 10 years respectively [14]. We have registered a same trend from 4 to 7 years of age, although from 7 to 9 years of age the growth of the girls´ feet increased to exceed the average length value that was registered in the boys.

As regards foot width, the average value that it was registered in our sample was generally higher in boys than in girls at all ages. Significant differences were found at 8 years of age, and from 10 to 12 years. Our results coincide with most observational studies that indicate that up to 8 years of age the foot grows predominantly in length, and after this age the width/length ratio in children is similar to that observed in adults [8,14,16,25].

We have found a correlation between body height and foot length at almost all ages. This correlation was strong at 4 and 12 years of age in boys and at 5, 9 and 11 years in girls. In our country, all the studies carried out in schoolchildren have shown a strong correlation between both parameters [14,22]. This strong correlation has also been observed in other studies carried out in schoolchildren in different ethnic groups [18,19].

Some authors establish that an increase in BMI leads to an increase in foot morphology particularly in metatarsal and heel width, but few correlational studies have analyzed these parameters in more detail. In children and adolescents, overweight seems to have an impact on foot structures, with changes in the anatomical structures, an abnormal distribution of the plantar pressure [22,23,26]. In our study we found a strong correlation between foot width and BMI at 12 years. In girls, we have also found a strong correlation between metatarsal width and BMI at 10 years of age. The correlation between BMI and foot length has also been poorly analyzed. In our study, we only found a weak correlation between BMI and foot length in boys at 10 years, and in girls at 11 years. Our results show a mild correlation between BMI and foot dimensions in length and width.

It is considered important to find indicators that allow identifying at what age the greatest growth of the foot occurs. The present study could be useful to determine at what stages it is necessary to control the size of the foot to prevent deformities due to footwear. The most useful finding of our study could be the strong correlation that was found between the growth of the foot in length and width, and the Tanner stage II in both sexes. We emphasize the importance of the periodic measurement of the longitudinal growth of the foot, and its utility for identifying the onset of the pubertal growth outbreak. We consider that dynamic shoe adjustments are needed, with greater attention in the age range of 7–11 years in girls, and 8–12 years in boys.

This type of cross-sectional studies allows the researcher to observe different anthropological parameters in schoolchildren, and to compare them with other ethnic cohorts. However, our study has some limitations to be acknowledged. We are aware of the need to carry out longitudinal studies that would allow analyzing the evolution of the anthropometric parameters until the end of the growth period. Finally, other cohorts should be studied to establish more valid findings. 

## 5. Conclusions

In this study it is strengthen the importance of the periodic measurement of the longitudinal growth of the feet since, together with other parameters, this parameter can be a valid index to establish the “maturation tempus” of schoolchildren. On the other hand, since a sudden increase in the growth of the feet can be considered as an early indicator of the onset of puberty, the routine measurement of the dimensions of the feet would allow identifying the onset of the pubertal growth spurt that occurs over a 5-year period (girls: 8–13 years, boys: 10–15 years). 

## Figures and Tables

**Figure 1 ijerph-18-04031-f001:**
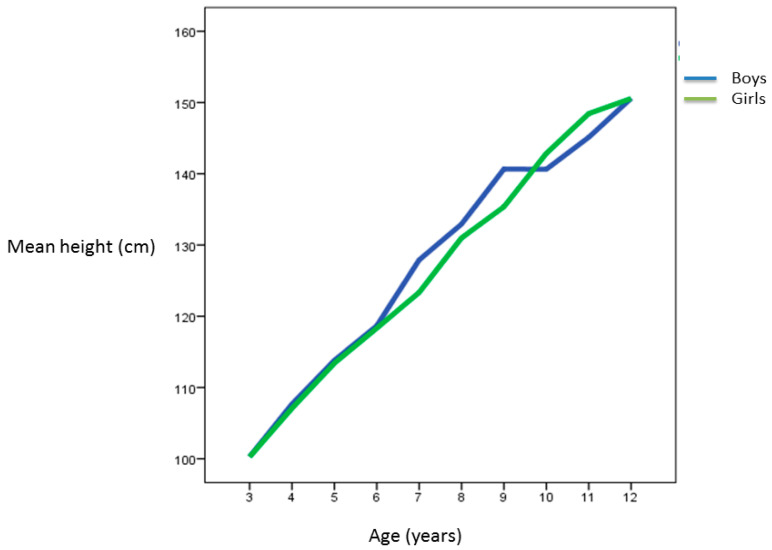
Height curve in relation to age and gender.

**Figure 2 ijerph-18-04031-f002:**
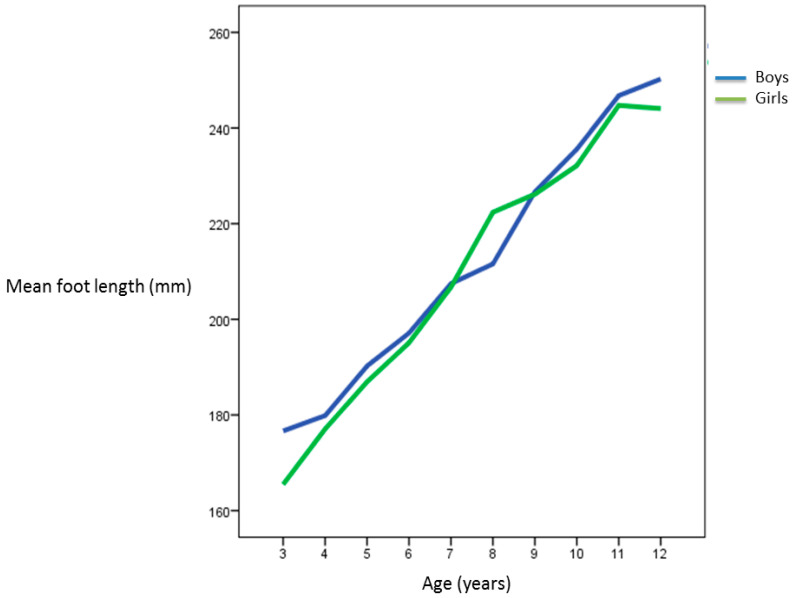
Foot length curve in relation to age and gender.

**Table 1 ijerph-18-04031-t001:** The physical features of the participants. Mean and standard deviation of anthropometric variables for different age groups in male and female groups.

**Boys**	**Height (cm)**	**Weight (Kg)**	**BMI**
3 years	100.29 (5.10)	16.11 (1.95)	1.11 (1.23)
4 years	107.68 (8.34)	18.30 (3.96)	15.52 (2.07)
5 years	113.81 (5.79)	21.65 (3.32)	17.23 (2.34)
6 years	118.67 (6.35)	23.52 (4.38)	16.52 (2.24)
7 years	127.92 (4.46)	30.00 (5.04)	18.04 (2.51)
8 years	132.94 (4.92)	31.83 (5.36)	18.00 (2.74)
9 years	140.67 (1.17)	35.81 (7.17)	18,86 (3.61)
10 years	140.64 (8.17)	38.73 (9.56)	19.18 (3.23)
11 years	145.12 (6.25)	41.64 (6.73)	19.84 (2.43)
12 years	150.60 (9.94)	43.80 (11.78)	19.00 (2.65)
**Girls**	**Height (cm)**	**Weight (Kg)**	**BMI**
3 years	100.26 (3.46)	16.79 (2.37)	16.86 (1.77)
4 years	107.9 (5.04)	18.79 (2.85)	16.35 (1.54)
5 years	113.4 (4.38)	21.08 (3.60)	16.69 (1.98)
6 years	118.33 (7.36)	24.07 (5.97)	17.01 (2.65)
7 years	123.36 (4.92)	25.82 (4.26)	17.04 (2.73)
8 years	131.00 (3.88)	32.64 (6.04)	19.12 (2.96)
9 years	135.38 (5.23)	32.04 (5.30)	17.54 (2.25)
10 years	142.86 (7.48)	40.69 (11.43)	19.72 (4.34)
11 years	148.45 (7.03)	43.70 (1.,55)	20.25 (3.42)
12 years	150.57 (9.9)	45.00 (8.76)	20.57 (3.46)

**Table 2 ijerph-18-04031-t002:** Mean, standard deviation and percentiles of foot anthropometric variables at different age groups and with the whole population.

**Boys**	**Foot Length**						**Foot Width**					
**Age**	**X ± SD (mm)**	**P_10_**	**P_25_**	**P_50_**	**P_75_**	**P_90_**	**X ± SD (cm)**	**P_10_**	**P_25_**	**P_50_**	**P_75_**	**P_90_**
3	176.66 (19.1)	156.6	165.75	173	182.25	206.7	6.60 (0.46)	6	6	6.88	7	7
4	179.85 (13.33)	165.6	170	180	188	200	7.05 (0.55)	6.2	7	7	7	8
5	190.23 (18.18)	173	180	190	196.75	201.4	7.29 (0.68)	6	7	7	8	8
6	197.12 (12.94)	180	192.25	200	203	209.7	7.47 (0.67)	6.2	7	8	8	8
7	207.46 (21.64)	172	203	210	216.5	229.2	7.95 (0.42)	7	8	8	8	8.5
8	211.59 (23.40)	167	200	219	229,5	233	7.89 (0.76)	7	7	8	8.55	9
9	226.66 (20.34)	207.8	220	229	241	247.2	8.46 (0.62)	7.45	8	8.5	9	9
10	235.57 (7.64)	217.6	226	236	246	257.9	8.71 (0.71)	8	8	9	9	10
11	246.79 (16.95)	225.5	233	247	260	270	8.82 (0.58)	8	8	9	9	9.5
12	250.25 (14.94)	232	234.75	250	266.75	-	9.17 (0.61)	8	9	9	9.75	-
**Girls**	**Foot Length**						**Foot Width**					
**Age**	**X ± SD (mm)**	**P_10_**	**P_25_**	**P_50_**	**P_75_**	**P_90_**	**X±SD (cm)**	**P_10_**	**P_25_**	**P_50_**	**P_75_**	**P_90_**
3	165.47 (10.01)	153	160	166	173.75	180	6.68 (0.69)	6	6	7	7	7.13
4	177.06 (12.89)	160.5	166	173	187	193	6.80 (0.49)	6	6.45	7	7	7.06
5	186.91 (11.10)	176.5	180	186	195.25	200.9	7.15 (0.64)	6	7	7	8	8
6	195.09 (19.03)	175.2	180	195	206	220	7.32 (0.70)	7	7	7	8	8
7	206.72 (20.15)	173	194	206	217.75	233	7.65 (0.81)	7	7	8	8	9
8	222.39 (24.72)	195.8	218	220	232.75	248.6	7.85 (0.69)	7	7	8	8.13	9
9	226.15 (11.13)	213	215	226	239.5	241.9	8.12 (0.52)	7.4	8	8	8.5	9
10	232.15 (15.11)	210.3	226	231	242.75	249.7	8.31 (0.71)	7	8	8	9	9
11	244.75 (12.41)	226.3	233.5	246	254.75	260	8.45 (0.53)	8	8	8	9	9
12	244.07 (12.98)	225	235	243	253	262.4	5.86 (0.82)	7	8	9	9	-

**Table 3 ijerph-18-04031-t003:** Correlation between the different variables in the male group.

Pearson Correlation (p)				
Boys	Foot Lenght-Height	Foot Lenght-BMI	Foot Width-Height	Foot Width-BMI
3 years	0.450 (0.016)	0.118 (0.548)	0.189 (0.336)	0375 (0.049)
4 years	0.719 (<0.001)	0.179 (0.245)	0.355 (0.018)	0.117 (0.448)
5 years	0.318 (0.038)	0.034 (0.827)	0.500 (0.001)	0.187 (0.231)
6 years	0.437 (0.029)	0.223 (0.284)	0.413 (0.040)	0.164 (0.433)
7 years	0.248 (0,232)	0.426 (0.034)	0.387 (0.056)	0.169 (0.418)
8 years	−0.224 (0.373)	−0.104 (0.680)	0.027 (0.915)	0.300 (0.226)
9 years	−0.003 (0.991)	−0.064 (0.784)	0.616 (0.003)	0.314 (0.165)
10 years	0.572 (0.005)	0.527 (0.012)	0.738 (<0.001)	0.697 (<0.001)
11 years	0.498 (0.011)	0.557 (0.004)	0.539 (0.005)	0.030 (0.887)
12 years	0.884 (0.047)	0.876 (0.051)	0.565 (0.321)	0.935 (0.020)

**Table 4 ijerph-18-04031-t004:** Correlation between the different variables in the female group.

Pearson Correlation (p)				
Girls	Foot Lenght-Height	Foot Lenght-BMI	Foot Width-Height	Foot Width-BMI
3 years	0.352 (0.139)	−0.085 (0.729)	0.019 (0.937)	0.128 (0.602)
4 years	0.167 (0.346)	0.287 (0.099)	0.264 (0.132)	0.267 (0.127)
5 years	0.524 (0.001)	0173 (0.312)	0.566 (<0.001)	−0.530 (0.001)
6 years	0.216 (0.280)	0.667 (<0.001)	0.719 (<0.001)	−0.202 (0.842)
7 years	0.345 (0.072)	0.190 (0.334)	0.471 (0.011)	−0.040 (0.842)
8 years	0.238 (0.252)	0.179 (0.391)	0.236 (0.256)	0.597 (0.002)
9 years	0.562 (0.004)	0.461 (0.023)	0.313 (0.136)	0.241 (0.256)
10 years	0.498 (0.006)	0.513 (0.004)	0.573 (<0.001)	0.812 (<0.001)
11 years	0.603 (0.005)	−0.070 (0.769)	0.543 (0.013)	0.184 (0.438)
12 years	0.563 (0.189)	0.116 (0.804)	0.212 (0.648)	0.569 (0.183)

## Data Availability

The data presented in this study are available on request from the corresponding author.

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
