# Peer review of "A Cross-Sectional Study of Foot Growth and Its Correlation with Anthropometric Parameters in a Representative Cohort of Schoolchildren from Southern Spain"

_ijerph, 2021, doi:10.3390/ijerph18084031_

Round 1

Reviewer 1 Report

The methodologies section is extremely limited.

Understanding the effort in data collection, the contribution is limited to showing comparative statistical analyzes, and numerical comparisons with previous works.

not presenting evidence of a novel contribution, where in the discussions or conclusions they comment on potential uses with useful impacts.

Author Response

1.- Material and method section has been conveniently expanded

2.- In discussion and conclusions sections novel contribution with useful impacts has been included

Reviewer 2 Report

Material and methods: neither the procedures, nor the recruitment, nor how the statistical analysis will be done are described;

They say "A previously validated measurement instrument was used to determine the maximum length and width in the foot [8]. It is not validated, it has only been used before.

No references to explain the foot measurement procedure

It is not correct to say: "the width of the foot with a computer program (ACCESS), using the X and Y coordinates of each point"

It remains to be explained how the Informed Consent Statement was obtained.

 mistakes:

Line 44. Space: adolescence [5] .Some studies

Line 71. 1,005 (sometimes dots and other commas: table 1: commas, and in the text of line 109, dots)

one point for the length, and two points for the width

Haight (table 1)

150.57 ± 9.9 better 150.57 (9.9)

It remains to be explained in Table 2 P10, P25, etc.

Figure 1 and 2 are not correct

References: They are not written according to the requirements of the journal

Author Response

1.    Precedures, recruitment and statistical analysis has been included in Materials and Methods section.
2.    The Foot measurements procedure and The inter-observer reliability of the researchers and the measurements has been included in measurements section
3.    The sentence  "the width of the foot with a computer program (ACCESS), using the X and Y coordinates of each point" has been modified to “The width was obtained from the X and Y 115 coordinates of each point using Access 11.0 (Microsoft Office 2010)”
4.    Mistakes has been corrected
5.    References have been written according to the requirements of the journal

Round 2

Reviewer 1 Report

the authors improved the work compared to the first version.

Integrating potential innovative uses is a challenge in every job and this one presents some interesting uses for the growth of the foot in schoolchildren.

Reviewer 2 Report

It is correct, now